# Metastasis-Directed Therapy in Oligometastatic Prostate Cancer: Biological Rationale and Systematic Review of Published Data

**DOI:** 10.3390/cancers17081256

**Published:** 2025-04-08

**Authors:** Francesco Fiorica, Teodoro Sava, Jacopo Giuliani, Umberto Tebano, Giuseppe Napoli, Antonella Franceschetto, Emilia Durante, Ilaria Campisi, Erica Palesandro, Fabio Turco, Consuelo Buttigliero, Fernando Munoz, Marcello Tucci

**Affiliations:** 1Department of Clinical Oncology, Section of Radiation Oncology and Nuclear Medicine, AULSS 9 Scaligera, 37122 Verona, Italy; umberto.tebano@aulss9.veneto.it (U.T.); giuseppe.napoli@aulss9.veneto.it (G.N.); antonella.franceschetto@aulss9.veneto.it (A.F.); 2Department of Clinical Oncology, Section of Medical Oncology, AULSS 9 Scaligera, 37045 Legnago, Italy; teodoro.sava@aulss9.veneto.it (T.S.); jacopo.giuliani@aulss9.veneto.it (J.G.); emilia.durante@aulss9.veneto.it (E.D.); 3Postgraduate School of Oncology, University of Turin, 10124 Turin, Italy; ilaria.campisi@unito.it; 4Oncology Unit, Agnelli Hospital, 10064 Pinerolo, Italy; erica.palesandro@aslto3.piemonte.it; 5Department of Oncology, Institute of Southern Switzerland, 6500CH Bellinzona, Switzerland; fabio.turco@eoc.ch; 6Department of Oncology, San Luigi Gonzaga Hospital, University of Turin, 10043 Orbassano, Italy; consuelo.buttigliero@unito.it; 7Radiation Oncology TomoTherapy Center, Hospital of Aosta, 11000 Aosta, Italy; fmunoz@ausl.vda.it; 8Department of Oncology, Cardinal Massaia Hospital, 14100 Asti, Italy; mtucci@asl.at.it

**Keywords:** Oligometastatic Prostate Cancer (OMPC), Metastasis-Directed Therapy (MDT), multimodal approach prostate cancer

## Abstract

Oligometastatic prostate cancer is an intermediate stage between localised and widespread disease, but the best treatment approach remains uncertain. While hormonal therapy is widely used, metastasis-directed therapy (MDT), such as targeted radiotherapy, may offer an alternative with fewer side effects. This review examines whether MDT provides better disease control, survival, and treatment-escalation outcomes alone or in combination with hormonal therapy. It also explores the biological and genetic factors that influence treatment response. By clarifying the role of MDT, this study aims to improve patient selection and guide future research toward more personalised treatment strategies.

## 1. Introduction

The definition of oligometastatic prostate cancer (OMPC) is based on a numerical criterion. However, identifying patients with fewer than five detectable metastatic lesions without fully considering the biological aggressiveness and systemic potential of the disease may lead to substantial overtreatment of some patients [1]. Indeed, introducing a metastasis-directed therapy (MDT) with a predominantly local effect could potentially cause side effects without leading to any survival advantage in patients affected by aggressive forms of prostate cancer with high risk of systemic spread [2].

There is an ongoing debate in the scientific and clinical community regarding whether OMPC represents a distinct biological state with unique molecular and clinical characteristics [3,4] or merely an early phase in the continuous spectrum of metastatic progression [5]. Some argue that OMPC is a transitional state in which the micrometastatic disease remains undetectable and that the visible lesions are just the first manifestations of a systemic process that will inevitably progress. Others suggest that OMPC exhibits more indolent biology that is potentially amenable to MDT, which could delay or even alter the course of metastatic disease that would be expected under systemic therapies alone [6]. MDT effectively targets and eliminates visible metastatic lesions without addressing micrometastatic disease or circulating tumour cells. Consequently, relying solely on locoregional therapy could result in the undertreatment of biologically aggressive forms of prostate cancer, allowing systemic disease to continue advancing. Conversely, in more indolent forms, aggressive systemic treatment might not add incremental benefit over MDT alone and potentially exposes patients to side effects.

In this work, we briefly describe the biological basis of oligometastatic spread in prostate cancer, providing a rationale for integrating MDT into treatment strategies for selected patients with oligometastatic disease. In addition, to investigate the current evidence regarding the role of MDT in OMPC patients and the potential additive role of systemic therapy in OMPC, we systematically reviewed the current literature on OMPC patients.

### 1.1. Biological Processes Characterising Oligometastatic Prostate Cancer (OMPC)

The concept of oligometastatic disease was introduced by Hellman and Weichselbaum [7] in 1995, challenging the traditional binary view of cancer as either localised or widely metastatic. Instead, they proposed that some tumours exist in an intermediate metastatic state, where the metastatic process is biologically constrained and may still be amenable to curative treatment. Although OMPC is clinically defined by the presence of fewer than five metastatic lesions [6], its biological underpinnings suggest a distinct tumour behaviour characterised by limited clonal evolution with fewer genetic mutations, a less aggressive tumour phenotype, and a unique tumour microenvironment (TME) that may restrict metastatic dissemination through immune surveillance.

### 1.2. Genomic and Molecular Landscape

In 2023, Sutera et al. [8] conducted a retrospective review of data on patients with biochemically recurrent or metastatic castration-sensitive prostate cancer (mCSPC) who had undergone somatic targeted sequencing. The study included 294 patients who were followed up for a median time of 58.3 months. Their findings revealed significant differences in the frequency of driver mutations across WNT signalling, cell cycle regulation, TP53, and the PI3K/AKT/mTOR pathway, depending on disease burden. Notably, patients with higher volumes of metastatic disease exhibited a higher incidence of these mutations, suggesting a correlation between genomic alterations and disease progression. Other research has investigated the clonal origin and dissemination of metastatic prostate cancer, emphasising the complexity of its clonal evolution.

Compared to metastatic disease, primary prostate cancer demonstrates lower mutation rates and reduced genomic instability. The gradual accumulation of genomic alterations is a key driver of tumorigenesis and metastatic progression, highlighting the evolutionary dynamics underlying disease advancement [9].

OMPC tumours harbour fewer high-risk mutations, particularly in genes such as TP53, PTEN, and RB1, which are frequently altered in widely metastatic prostate cancer. A study analysing high-risk mutational signatures (ATM, BRCA1/2, RB1, TP53) found that 74.3% of OMPC patients did not exhibit these alterations, suggesting a less aggressive genetic profile [10].

While definitive evidence is lacking, the classification of OMPC as a low-volume metastatic disease suggests that its tumour-cell populations may be more homogeneous and less prone to further metastatic dissemination compared to those in polymetastatic disease [11,12,13]. The relative genomic stability and limited clonal evolution observed in OMPC imply a less aggressive metastatic potential, supporting the idea that OMPC represents a biologically distinct entity rather than an early phase of widespread metastasis.

As reported by Sutera et al. [14], OMPC tumours are characterized by a transcriptomic profile indicating a higher androgen response. This increased AR dependence suggests that OMPC may be more responsive to androgen-deprivation therapy (ADT) compared to polymetastatic prostate cancer, where AR-independent mechanisms and resistance pathways frequently emerge [15]. Observations of a more favourable clinical trajectory in OMPC reinforces the hypothesis that tumour biology, rather than just the timing of metastasis, plays a key role in disease progression.

Similarly, the PI3K/AKT/mTOR pathway, a critical regulator of tumour survival, proliferation, and therapy resistance [16,17], appears less dysregulated in OMPC than in polymetastatic disease. This pathway is frequently altered in advanced prostate cancer [18], contributing to tumour progression, metabolic adaptation, and resistance to ADT. However, in OMPC, fewer genetic mutations and structural alterations have been observed in key PI3K/AKT/mTOR cascade regulators [10], suggesting a lower degree of pathway activation. This relative preservation of pathway integrity may indicate that OMPC tumours rely more on AR signalling than on alternative oncogenic pathways for survival, making them more susceptible to AR-targeted therapies and potentially less prone to early treatment resistance.

Another key biological factor influencing metastatic potential is the epithelial−mesenchymal transition (EMT), which enables tumour cells to lose their epithelial characteristics, gain motility, and invade distant tissues. EMT is a hallmark of aggressive metastatic progression, allowing tumour cells to evade immune surveillance, enter the circulation, and establish new metastatic colonies [19]. In polymetastatic prostate cancer, EMT activation is widespread [20], driven by factors such as hypoxia, inflammatory cytokines, and TGF-β signalling, and leads to increased metastatic dissemination and therapy resistance. In contrast, OMPC exhibits reduced EMT activity, resulting in lower tumour-cell plasticity and a decreased ability to colonise distant sites [21]. The limited EMT activation in OMPC may result from less genomic instability, a more differentiated tumour phenotype, and a more immunologically active tumour microenvironment, which collectively act as barriers to systemic spread.

### 1.3. MicroRNA and Epigenetic Regulation

MicroRNAs play a crucial role in the regulation of gene expression and tumour behaviour. Variations in their expression profiles may explain why OMPC remains localised while polymetastatic disease spreads aggressively. In oligometastatic patients, Lussier et al. [22] identified a distinct molecular signature, including miR-200c, that could differentiate individuals who would go on to experience widespread metastatic progression (defined as five or more new metastases within four months or dissemination within a body cavity) from those who remained in an oligometastatic state. Experimental research indicates that epigenetic modulation of the epithelial−mesenchymal transition (EMT) is a key factor in defining this phenotype, offering more profound insights into the cellular mechanisms that drive distinct metastatic pathways [21]. These findings hold significant translational potential, as clinical biomarkers reflecting intermediate EMT states could improve prognostic stratification in metastatic patients and support more tailored treatment approaches. For instance, patients with tumours exhibiting a predominantly “highly mesenchymal” profile—tumours that are more likely to remain oligometastatic—may respond better to early local therapies. Conversely, those with a “quasi-mesenchymal” phenotype, one associated with a higher risk of widespread metastasis, may benefit more from early systemic treatments [23].

### 1.4. Tumor Microenvironment (TME) and Immune Response

The TME in OMPC is critical in limiting disease progression and shaping metastatic potential. Compared to polymetastatic prostate cancer, OMPC is associated with a more immunologically active and restrictive microenvironment, which may contribute to its limited metastatic spread [24]. OMPC is associated with a more immunologically active microenvironment characterised by higher levels of tumour-infiltrating lymphocytes (TILs) and natural killer (NK) cells, which suggests a more robust immune-surveillance mechanism.

Additionally, stromal interactions with fibroblasts and the extracellular matrix (ECM) limit cancer-cell invasiveness, further contributing to OMPC’s constrained metastatic behaviour [25,26]. Another key observation is that OMPC patients have lower levels of circulating tumour cells (CTCs) and circulating tumour DNA (ctDNA), indicating reduced systemic dissemination compared to patients with widespread metastases [27].

A significant biological feature of OMPC is tumour dormancy and immune surveillance. Disseminated tumour cells (DTCs) in OMPC may remain dormant, restrained by immune mechanisms, and lack the necessary signals to establish new metastatic sites. This state of biological restraint may delay disease progression, creating a therapeutic window in which metastasis-directed treatments such as stereotactic body radiotherapy (SBRT) or metastasectomy can effectively eliminate metastatic sites before further dissemination occurs.

## 2. Methods

### 2.1. Study Selection

MEDLINE/PUBMED and EMBASE searches were performed to identify eligible reports, which were those published up to December 2024 evaluating metastasis-directed therapy in managing mPC patients. The proceedings of the European Society for Radiotherapy and Oncology, European Society of Medical Oncology, American Society for Radiation Oncology, American Society of Clinical Oncology, European Uro-Oncology Group, and American Urological Association annual meetings were examined for presented abstracts. Studies were included if they analysed at least ten patients, regardless of whether they were comparative randomised, nonrandomised, or single-arm studies and if progression-free survival and/or overall survival were analysed and reported as endpoints.

### 2.2. Data Extraction

Data extraction was conducted independently by three investigators (GN, UT, AF), following the Preferred Reporting Items for Systematic Reviews and Meta-Analyses (PRISMA) guidance. This review was registered as PROSPERO 1007633. For each study, the following information was extracted: publication or presentation date, first author’s last name, sample size, primary endpoints, regimens used, follow-up period, number of outcome events (progression-free survival, overall survival, androgen-deprivation-free survival), study design, subgroup analyses, and toxicities. The primary endpoint of this study was the proportion of men not requiring treatment escalation after MDT, including ADT for castration-sensitive mPC and hormonal therapy or chemotherapy for castration-resistant mPC. Secondarily, progression-free survival and overall survival were calculated as a pooled result.

### 2.3. Statistical Methods

The methodological quality of the studies was assessed using a checklist for quality appraisal of case-series studies produced by the Institute of Health Economics (IHE), which was modified to improve applicability [28].

In clinical trials with a time-dependent outcome (death or disease recurrence), survival curves were used to describe the risk of the event over time. The most informative finding was a summary survival curve used for meta-analyses of studies reporting a survival curve. We used the nonparametric approach of Combescure et al. [29] to assess pooled survival probabilities from several single-arm studies. This approach is a version of the aggregated data method applied to the product-limit estimator of survival and uses random effects to model between-study heterogeneity. The between-study covariance matrix was estimated using the multivariate extension of DerSimonian and Laird’s method [30,31]. This approach has several advantages compared to meta-analyses of survival probabilities at a single time point [32]. First, estimating the pooled survival probability at time *t* also incorporates all studies that ended before *t* because these studies contribute to the estimated conditional survival probabilities for time intervals before *t*. Second, this approach does not require assumptions about the shape of survival curves. Finally, the pooled survival probabilities are guaranteed not to increase over time. For all analyses, a *p*-value <0.05 was considered statistically significant. All analyses and graphics were completed in the R Statistical Computing Environment (R Foundation for Statistical Computing, Vienna, Austria).

## 3. Results

Eighteen studies published between 2014 and 2024 met the inclusion criteria and were selected for the analysis. Most studies were conducted in Europe (*n* = 8) or North America (*n* = 6), with the rest being from Australia (*n* = 3) and Asia (*n* = 1) (Figure 1).

All studies were prospective, and three were randomised trials [33,34,35]. The selected studies included a total of 1058 patients treated with metastasis-directed radiotherapy, although the number of enrolled patients varied (range 29 to 199) [36,37]. The characteristics of the patients included are summarised in Table 1. Five hundred twenty-one patients (55.4%) had a Gleason score equal to or less than 7; 406 (43.2%) had a Gleason score > 8. The median PSA value was 4.3 ng/mL (0.61–10.2). Seventy-four per cent of treated patients were castration-sensitive, and the remaining 26% were castration-resistant, as principally analysed by five studies [35,38,39,40,41]. Positron emission tomography with a prostate cancer-specific tracer was used for staging purposes in 88.4% (n = 90) of patients. Ninety-four per cent (n = 923) had three or fewer metastases. Nodal-only metastases were present in 49.7% (n = 520), while bone-only disease occurred in 43.8% (n = 459). Both sites of metastases were observed in 5.7% of included patients (n = 60). Median follow-up from MDT was 24 months [33,36,37,41,42,43,44,45,46,47,48,49,50].

### 3.1. Treatment Escalation

Data on the need for treatment escalation were available for all studies. Three hundred twenty-two patients had castration-resistant disease. The remaining 736 patients with castration-sensitive disease were treated with two treatment approaches: MDT delivered as a single treatment (in 415 patients) and hormone-suppressive therapy (in 321 patients). Treatment escalation is defined as starting a new treatment program for disease progression, including ADT in systemic-treatment-naïve patients and second-line hormonal therapy or chemotherapy for patients treated with a multimodal approach. Patient characteristics, including the number and site of metastasis, Gleason score, median PSA value, and age, were similar in these three groups.

In hormone-naïve patients, in seven studies, the estimated 2 year treatment-escalation-free survival rate was 61.9% (CI: 54.9–69.6%), with pooled survival rates ranging from 76.5% at 1 year to 38.6% at 4 years. The median time to treatment escalation was 32.7 months. Low heterogeneity was identified (i^2^ = 17.9%). Figure 2a shows treatment-escalation-free survival curves extracted from studies and the summary curve for patients treated with MDT alone.

In the group treated with metastasis-directed therapy and hormone manipulation (n = 321), the estimated 2 year treatment-escalation-free survival rate was 68.7% (CI: 51.1.6–92.4%), with a pooled survival rate ranging from 87.1% at 1 year to 39.4% at 4 years. The median time to treatment escalation was 38.6 months. Heterogeneity was low (i^2^ = 18.5%). Figure 2b shows the disease-control survival curves extracted from studies and the summary curve for patients treated with a multimodal approach.

When the pooled 2 year rates of two groups, MDT (61.9%) vs. MDT and hormone-suppressive therapy (68.7%), were compared, no significant difference in treatment-escalation-free survival was found (*p* = 0.06).

The estimated treatment-escalation-free survival rate in the castration-resistant group (n = 322) was 63.3% (CI: 39.5–84.4). Heterogeneity between studies was low (i^2^ = 28.1%). Figure 2c shows the treatment-escalation-free survival curves extracted from studies and the summary curve for CRPC patients. The median time to treatment escalation or progression was 30.9 months.

### 3.2. Overall Survival

The OS curve using individual patient (IP) survival data of 486 patients is reported in Figure 3a. The median OS was 112.6 months. The pooled estimate of the 2 year survival rate was 90.6% (range 85.1–96.5%), while the 4 year and 10 year rates were 80.1% (range 72.4–88.8%) and 44.1% (range 29.9–64.9%), respectively. There is low heterogeneity among studies (i^2^ = 35.2%). A subgroup analysis was used to identify potential sources of heterogeneity among the studies. When data from the hormone-sensitive group (268 patients from 7 studies) were pooled, the estimated 2 year and 4 year survival rates were 94.8% (range 91–98.8%) and 84.8% (range 78.2–91.9%), respectively. The studies had low heterogeneity (i^2^ = 10.4%). The median OS calculated in this group was 116.2 months (Figure 3b).

We also investigated whether there were OS differences among patients with castration-sensitive disease between those treated with MDT alone exclusively and those who received MDT and ADT.

OS data for patients with castration-sensitive disease treated with MDT alone were reported in three studies including a total of 117 patients. The estimated 2 year and 4 year overall survival rates were 96.4% (95% confidence interval [CI], 92.9–100%) and 89.1% (95% CI, 82.3–96.5%), respectively. The heterogeneity was very low (i^2^ = 3.4). The median OS was not reached. Three studies reported OS in patients with castration-sensitive disease treated with MDT combined with ADT, for a total of 121 patients. The estimated 2 year and 4 year overall survival rates were 86.1% (95% CI 79.2–93.7%) and 74.8% (95% CI 64.6.3–86.5%), respectively. Heterogeneity was low (i^2^ = 15.4%).

When these two groups were compared, differences in OS rate were observed at 2 and 4 years (*p* = 0.004).

On analysis of the CRPC group (183 patients in 4 studies), the estimated 2 year and 4 year survival rates were 80.6% (range 72.2–90.1%) and 65.9% (range 38.9–90%), respectively. The studies had very low heterogeneity (i^2^ = 7.8%). The median OS in this group was not reached (Figure 3c).

### 3.3. Disease-Progression-Free Survival

Figure 4a reports the disease-progression-free survival (DPFS) curve using IP DPFS data for 660 patients from 14 studies. The median DPFS calculated in this way was 25.9 months. The pooled estimates of the 2 year and 4 year DPFS rates were 52.7% (range 40.2–69.2%) and 28.4% (range 15.7–51.5%), respectively, although high heterogeneity among studies was found (i^2^ = 52.1%). Subgroup analyses were used to identify potential sources of heterogeneity among the studies.

When data from 12 studies including patients with castration-sensitive disease were pooled, the estimated 2 year and 4 year DPFS rates were 51.9% (range 36.9.2–71.1%) and 27.1% (range 14.2–51.7%), respectively. The studies had moderate-to-high heterogeneity (i^2^ = 49.5%). The median DFS time calculated in this group was 24.8 months.

To explore heterogeneity in patients with castration-sensitive disease, we analysed the two patient groups: those treated with MDT alone and those who received a combination of MDT and hormone suppression.

Seven studies reported DPFS data for patients with castration-sensitive disease (n = 285) treated with MDT alone (Figure 4b). The estimated 2 year and 4 year DPFS rates were 31.1% (range 23.2–41.7%) and 18% (range 11.8–27.8%), respectively. Heterogeneity between studies was low (i^2^ = 15.5%). By contrast, the estimated 2 year and 4 year DPFS rates were 50.5% (range 34.4–74.2%) and 27.9% (range 15.8–49.4%) for patients treated with MDT and ADT (7 studies, 267 patients). However, heterogeneity between studies was high (i^2^ = 66.8%). Given these limitations, when the two groups were compared, the data showed that the combination approach significantly increases both 2 year and 4 year DPFS rates (*p*-value <0.00001 and 0.006, respectively).

Among the five studies that included CRPC patients (n = 129, Figure 4c), the estimated 2 year DFFS rate was 49.8% (range 16.2–77.1%). The studies had very low heterogeneity (i^2^ = 3.8%). The median DPFS calculated in this group was 23.8 months.

## 4. Discussion

Understanding the molecular and cellular mechanisms underlying the progression of OMPC is essential for refining precision-medicine approaches and optimising therapeutic strategies in this subset of metastatic disease. While the numerical definition (≤5 metastases) provides a clinical framework, genomic profiling is crucial for tailoring treatment strategies. The key clinical question remains whether MDT alone is sufficient or whether it must be combined with systemic treatment.

From a biological point of view, MDT may disrupt the metastatic cascade, preventing further dissemination of tumour cells and altering the TME. Local control of metastatic sites may reduce the release of ctDNA, which are critical mediators of disease progression and metastasis formation [51]. Additionally, MDT may modulate the immune microenvironment by inducing immunogenic cell death, enhancing antigen presentation, and potentially synergising with the immunomodulatory effects of hormonotherapy in the early phase. ADT offers effective control of systemic disease in the short term, increasing TILs and upregulating immune checkpoints [52]. Conversely, prolonged ADT use may disrupt the tumour microenvironment and impair immune surveillance, contributing to immune evasion and tumour persistence [53].

Our work shows that MDT alone is associated with long-term survival and extended treatment-free intervals in OMPC, challenging the notion that the addition of routine hormonotherapy provides clear benefits. While the MDT + ADT approach significantly improves DPFS, this improvement does not translate into OS or TEFS advantages, raising questions about the necessity of systemic therapy in all patients.

The observed DPFS benefit with MDT + ADT (*p* < 0.00001 for 2 years and 0.006 for 4 years) may reflect a biological interaction between local and systemic treatments. Hormonotherapy suppresses the AR signalling essential for survival of prostate cancer cells and acts on both local and systemic disease, while MDT eradicates known metastatic sites but does not address micrometastatic diseases [50]. Hormonotherapy also limits cancer cells’ ability to disseminate and establish new metastatic sites by inhibiting EMT and reducing tumour invasiveness [54]. This effect complements MDT by preventing new metastatic spread, further contributing to prolonged DPFS. The combination treatment delays disease progression by targeting macroscopic and microscopic tumour components, significantly extending DPFS compared to MDT alone.

However, the lack of a survival advantage suggests that delaying disease progression does not necessarily improve long-term outcomes, as patients eventually transition to systemic therapy at the time of progression. Given that treatment-escalation-free survival remains unchanged, this reinforces the concept that immediate systemic therapy upon progression may not be required in many cases and that delaying its initiation does not appear to compromise outcomes.

OMPC is a distinct pathological entity following a unique biological and clinical trajectory. OMPC patients often have a less aggressive neoplastic disease that progresses more slowly than polymetastatic disease [11,12,13].

Despite the superior disease control achieved with MDT + hormonotherapy, the benefit does not extend to OS. The observed absence of effects on OS with combination treatment compared to MDT may be due to the selection of patients in studies not planned to evaluate OS differences. Patients treated with MDT alone often have less aggressive disease compared with those treated with combination therapy.

Alternatively, we can hypothesise a negative impact of hormonotherapy on outcome in some cases of OMPC. Our results may suggest that the long-term systemic toxicities of ADT may negatively impact survival outcomes. ADT is associated with cardiovascular complications, metabolic syndrome, osteoporosis, and increased diabetes risk, which can offset its oncological benefits, particularly in low-burden disease [55]. Patients receiving MDT alone avoid systemic toxicities, maintaining better overall health, which may translate into improved OS.

Another factor contributing to this survival benefit is MDT’s immunogenic effects. In particular, stereotactic body radiotherapy (SBRT) can induce immunogenic cell death, releasing tumour antigens and triggering a systemic anti-tumour immune response. This can result in abscopal effects, where untreated metastases regress due to enhanced immune activation [47,48]. While ADT initially enhances immune infiltration, long-term use may have immunosuppressive effects, impairing the body’s natural ability to control residual disease [56]. Thus, MDT alone may better preserve immune function, leading to superior OS.

In addition, hormonotherapy can induce the clonal selection of androgen-independent cells. Hormonotherapy exerts selective pressure, favouring the survival and proliferation of aggressive, androgen-independent clones, accelerating the progression to CRPC. MDT alone directly eliminates metastatic lesions without exerting such selective pressure, potentially delaying the emergence of resistant tumour subpopulations [57]. Moreover, patients treated with MDT alone maintain a higher quality of life, better preserving physical function, metabolic health, and psychological well-being, thus enhancing adherence to follow-up care and improving long-term survival.

Given these observations, a personalised approach is necessary to determine which patients require MDT alone versus MDT + systemic therapy. Not all OMPC patients benefit equally from systemic therapy. Genomic stratification and molecular biomarkers can refine treatment selection as follows:Low mutational burden (74.3% of OMPC cases) → Better prognosis. MDT alone may be sufficient.High-risk mutations (e.g., BRCA1/2, TP53, ATM) → MDT + systemic therapy is likely required to prevent early progression.

Liquid biopsy is emerging as a key tool for detecting genomic alterations and identifying OMPC patients at higher risk of early progression. Colosini et al. [58] demonstrated that liquid-biopsy-based molecular profiling can stratify OMPC patients and predict SBRT response. Their study found that patients with BRCA1 mutations had higher rates of SBRT failure, highlighting the importance of real-time molecular profiling in individualising MDT strategies.

In CRPC patients (*n* = 322), MDT shows a treatment-escalation-free survival rate of 63.3% at 2 years and a median progression-free period of 30.9 months. This interesting period of disease control translates into improved control of clinical disease progression, with a 2 year control rate of 49.8% and a median of 23.8 months. Notably, OS rates also show promising outcomes, with a 2 year survival rate of 80.6% and a 4 year survival rate of 65.9%. MDT could also reduce the overall metastatic burden in CPRC patients, potentially altering the natural progression of the disease. It can eliminate dominant metastatic clones to prevent further dissemination [38], modulate the tumour microenvironment, delay the onset of widespread resistance, and enhance the immune response [59].

## 5. Limitations

While the study provides valuable insights into the role of MDT in OMPC, several limitations must be considered. The analysis is based on a systematic review and meta-analysis, incorporating multiple studies with different methodologies, patient populations, and follow-up durations. Variability in inclusion criteria, imaging techniques, and treatment protocols may introduce bias and limit the generalizability of findings. Many included studies are single-arm phase II trials, limiting the authors’ ability to establish causal relationships between MDT and improved outcomes. There are few large phase III RCTs comparing MDT + systemic therapy versus systemic therapy alone in OMPC patients. This study includes different MDT approaches, which makes direct comparisons difficult. The optimal combination of MDT with systemic therapy remains uncertain.

Furthermore, studies often include highly selected patients with low disease burden, good performance status, and favourable prognostic factors, who may not represent the broader OMPC population. Patients with higher-risk genomic profiles might not respond as well to MDT, but stratification based on molecular markers is often lacking. Future trials should integrate biomarker-based approaches to refine treatment decisions and minimise overtreatment. While MDT is a promising strategy for OMPC, its role requires further validation in large, randomised trials with biomarker-driven patient selection. Future research should focus on standardising MDT protocols, optimising combinations with systemic therapies, and improving assessment of long-term outcomes.

## 6. Conclusions

Metastasis-directed therapy (MDT) represents a promising approach for managing oligometastatic prostate cancer (OMPC). It can potentially delay disease progression and reduce the need for immediate systemic therapy. This systematic review highlights the finding that MDT alone can offer promising short- and long-term outcomes in OMPC, with no clear survival advantage from the routine addition of ADT.

While MDT + ADT improves DPFS, its impact on overall survival and treatment-escalation-free survival remains uncertain. Therefore, MDT alone represents a valuable and effective approach for many patients, and the decision to incorporate ADT should be carefully individualised based on patient-specific biological and genomic characteristics.

Future research should focus on biomarker-driven approaches to refine treatment selection, ensuring that MDT is applied where it offers the greatest benefit while minimising unnecessary systemic exposure. Integrating liquid biopsy, genomic profiling, and tumour-microenvironment analysis may enhance personalised treatment strategies, optimising outcomes for patients with OMPC. Large-scale, prospective trials are needed to establish the most effective treatment combinations and define the long-term role of MDT in the evolving landscape of prostate cancer therapy.

## Figures and Tables

**Figure 1 cancers-17-01256-f001:**
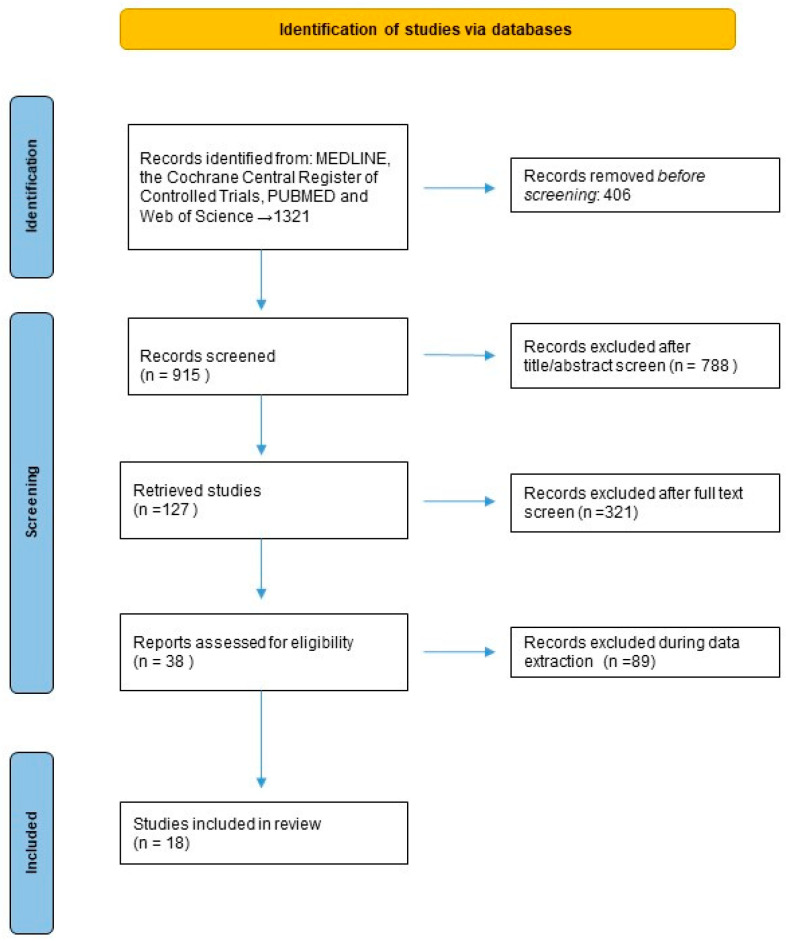
Study flow chart.

**Figure 2 cancers-17-01256-f002:**
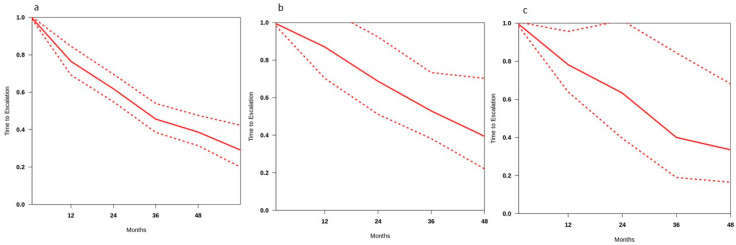
Treatment-escalation-free survival rates in (**a**) patients treated with MDT alone, (**b**) patients with castration-sensitive disease treated with MDT and androgen-suppression therapy, and (**c**) patients with castration-resistant disease.

**Figure 3 cancers-17-01256-f003:**
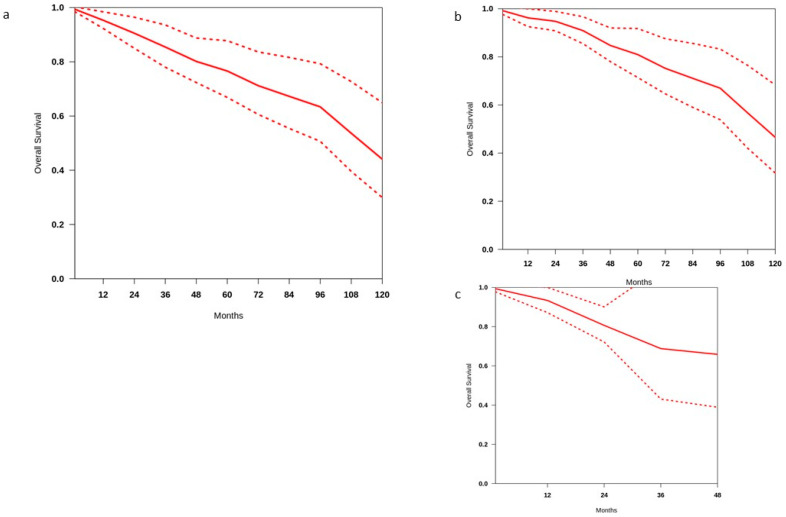
Overall survival in (**a**) all patients treated with MDT, (**b**) patients with castration-sensitive disease, and (**c**) patients with castration-resistant disease.

**Figure 4 cancers-17-01256-f004:**
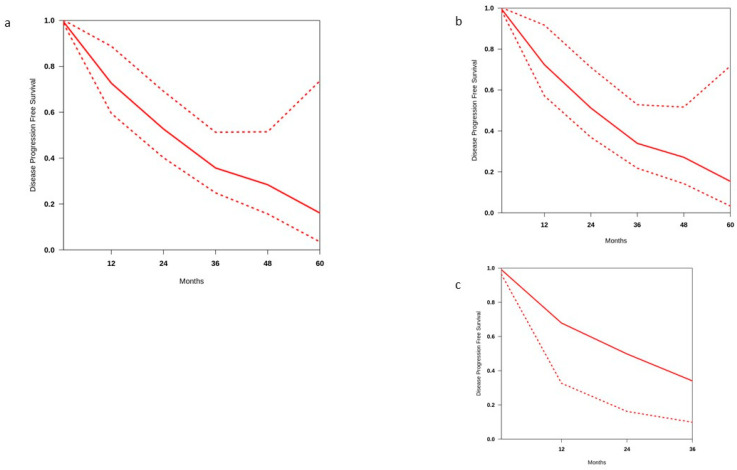
Disease-progression-free survival in (**a**) all patients treated with MDT, (**b**) patients with castration-sensitive disease, and (**c**) patients with castration-resistant disease.

**Table 1 cancers-17-01256-t001:** Summary of studies on metastasis-directed therapy (MDT) in prostate cancer.

Author (Reference)	N	Mean Age	Gleason Score (≤7/≥8)	PSA Median	PET Scan (Yes/No)	Prior Surgery	Prior RT	ADT (Yes/No)	Castration Sensitivity (Sensitive/Resistant)	Number of Metastases (≤3/>3)	Site of Metastases (Bone/Node/Both/Other)	Time from Primary Therapy to SBRT (Months)
Bowden et al. [37]	199	67.4	115/77	1.8	138/38	185	12	33/152	185/14	165/34	45/126/24/4	3.8
Decaestecker et al. [42]	50	59	17/33	5.1	18/32	42	8	0/50	50/0	50/0	22/24/2/1	5.3
Deodato et al. [43]	37	73.5	20/17	1.8	nr	nr	nr	37/0	37/0	37/0	50/0/0/0	Nr
Evans et al. [44]	37	67.6	21/16	4.4	nr	Nr	27	5/25	0/25	12/37	36/1/0/0	5.3
Glickman et al. [45]	74	61	58/14	1	74/0	74	0	0/74	74/0	62/12	9/64/1/0	4.9
Gomez-Iturriaga et al. [46]	49	71	25/24	4.3	49/0	32	14	40/9	49/0	48/1	13/34/2/0	nr
Hao et al. [36]	29	67	16/12	9.7	nr	nr	13	1/29	0/29	0/27	16/8/5/0	0.7
Holsher et al. [47]	63	72	44/18	10.2	63/0	60	3	0/63	63/0	61/2	16/43/4/0	4.7
Kneebone et al. [48]	57	64	38/19	2.12	57/0	50	7	57/0	57/0	57/0	18/37/2/0	5.6
Ost et al. [33]	31	62	21/10	5.3	31/0	24	7	0/31	31/0	31/0	14/17/0/1	5.3
Phillips et al. [34]	36	68	25/11	6	36/0	30	6	0/36	36/0	NR/NR	15/21/0/0	1.8
Supiot et al. [49]	67	67.5	57/10	3.7	67/0	61	6	67/0	67/0	62/5	0/67/0/0	4.5
Siva et al. [50]	39	70	18/21	6.4	39/0	18	15	33/0	33/6	39/0	21/13/2/0	0.9
Deek et al. [38]	68	nr	24/42	8.8	nr	nr	nr	68/0	0/68	nr	nr	nr
Francolini et al. [35]	75	74	13/62	3.4	71/4	nr	nr	75/0	0/75	75/0	42/33/0/0	nr
Phillips et al. [34]	89	70.8	nr	nr	89/0	74	nr	89/0	0/89	89/0	71/5/12/1	nr
Pan et al. [40]	29	69	9/20	0.61	29/0	28	1	29/0	0/29	26/3	15/10/4/0	2.9
Pasqualetti et al. [41]	29	69.9	nr	3.43	29/0	nr	nr	29/0	0/29	nr	13/16/1/0	3.2

nr = not reported, ADT = androgen-deprivation therapy, SBRT = stereotactic body radiotherapy.

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
