# Peer review of "Metastasis-Directed Therapy in Oligometastatic Prostate Cancer: Biological Rationale and Systematic Review of Published Data"

_cancers, 2025, doi:10.3390/cancers17081256_

Round 1
Reviewer 1 Report
Comments and Suggestions for Authors
Fiorica et al have performed a systemic review of metastasis-directed therapy in oligometastatic prostate cancer. The review provides an excellent introduction and background for the disease state. They identify the studies that have been performed of MDT with or without systemic therapy in both HSPC and CRPC states. Important outcomes are highlighted including survival, time to progression and time to needing additional therapy. Discussion and conclusions regarding the precise role of ADT can be improved however.
The paper's discussion and conclusions are written to support a conclusion that there is value in ADT along with MDT. However, this is not what the data from the study supports nor would be supported a priori from prior known data in biochemical recurrence only. MDT is associated with long term survival and treatment-free intervals:
Overall survival: significantly higher in MDT alone compared to MDT + ADT
Treatment escalation free survival: no significant differences for MDT alone vs MDT + ADT
Disease progression free survival: significantly improved with MDT+ ADT compared to MDT alone. However, as noted in the paper, this did not necessarily immediately require further therapy and did not translate to any other hard improvements in outcomes.
Therefore, based upon the data, the best conclusion is that there is substantial evidence for MDT alone with excellent short and long term outcomes. Furthermore, the addition of ADT to MDT is not associated with improvement in survival (in fact had inferior survival as discussed) and should generally not be used absent further data of benefit. It is reasonable to suggest that there would be a subset of patients wherein a progression of disease event is worth treatment with ADT, but that would be the exception.
Despite this being the data, the abstract highlights the potential benefit of combined therapy. The discussion also starts with the theoretical benefits of combined therapy. however, I don't think the data support this and would recommend a revision to reflect that.
Author Response
Reviewer #1:
Fiorica et al have performed a systemic review of metastasis-directed therapy in oligometastatic prostate cancer. The review provides an excellent introduction and background for the disease state. They identify the studies that have been performed of MDT with or without systemic therapy in both HSPC and CRPC states. Important outcomes are highlighted including survival, time to progression and time to needing additional therapy.
We thank you for what is written.
The paper's discussion and conclusions are written to support a conclusion that there is value in ADT along with MDT. However, this is not what the data from the study supports nor would be supported a priori from prior known data in biochemical recurrence only. MDT is associated with long term survival and treatment-free intervals: Overall survival: significantly higher in MDT alone compared to MDT + ADT, Treatment escalation free survival: no significant differences for MDT alone vs MDT + ADT, Disease progression free survival: significantly improved with MDT+ ADT compared to MDT alone. However, as noted in the paper, this did not necessarily immediately require further therapy and did not translate to any other hard improvements in outcomes.
We appreciate your constructive input and thank the reviewer for the comment.
We have revised the abstract, discussion, and conclusions to ensure our findings align more accurately with the presented data.
We have refined the abstract's conclusion to state that “MDT alone offers promising outcomes in OMPC and represents a valuable, valid, and often preferable strategy. Combined with ADT, it significantly improves disease progression-free survival, but its impact on overall survival remains uncertain. Given these findings, the decision to incorporate ADT should be tailored to individual patient characteristics and clinical context.”
In the discussion, we have clarified that MDT alone is associated with long-term survival and extended treatment-free intervals, challenging the assumption that the routine addition of ADT provides clear benefits. “Our work shows that MDT alone is associated with long-term survival and extended treatment-free intervals in OMPC, challenging the notion that routine addition of hormonotherapy provides clear benefits. While MDT+ADT significantly improves DPFS, this improvement does not translate into OS or TEFS advantages, raising questions about the necessity of systemic therapy in all patients. The observed DPFS benefit with MDT+ADT (p <0.00001 for 2 years and 0.006 for 4 years) may reflect a biological interaction between local and systemic treatments. …………………..However, the lack of a survival advantage suggests that delaying disease progression does not necessarily improve long-term outcomes, as patients eventually transition to systemic therapy at the time of progression. Given that treatment escalation-free survival remains unchanged, this reinforces the concept that immediate systemic therapy upon progression may not be required in many cases, and delaying its initiation does not appear to compromise outcomes”.
The revised conclusion now emphasizes that MDT alone is an effective treatment strategy for many OMPC patients. “ MDT alone can offer promising short- and long-term outcomes in OMPC, with no clear survival advantage from the routine addition of ADT. While MDT + ADT improves DPFS, its impact on overall survival and treatment escalation-free survival remains uncertain. Therefore, MDT alone represents a valuable and effective approach for many patients, and the decision to incorporate ADT should be carefully individualized based on patient-specific biological and genomic characteristics”.
These revisions ensure that our conclusions align with the data while providing a balanced interpretation of MDT and MDT + ADT in OMPC.
Therefore, based on the data, the best conclusion is that there is substantial evidence for MDT alone with excellent short—and long-term outcomes. Furthermore, the addition of ADT to MDT is not associated with improvement in survival (in fact, it had inferior survival, as discussed) and should generally not be used absent further data of benefit. It is reasonable to suggest that there would be a subset of patients wherein a progression of disease event is worth treatment with ADT, but that would be the exception.
Despite this being the data, the abstract highlights the potential benefit of combined therapy. The discussion also starts with the theoretical benefits of combined therapy. however, I don't think the data support this and would recommend a revision to reflect that.
We appreciate your critical analysis and have carefully revised the manuscript to ensure the conclusions fully align with the presented data.
We have revised the abstract to reflect that MDT alone offers promising outcomes in OMPC. We have emphasized that MDT alone is a valuable, valid and preferable approach and that ADT should be considered individually rather than as a routine addition.
We acknowledge that the previous discussion may have emphasized the theoretical benefits of combined therapy. To address this, we have clarified that while MDT+ADT improves DPFS, this does not translate into a survival advantage or a reduced need for subsequent treatment escalation. The revised discussion now emphasizes that MDT alone is associated with long-term survival and treatment-free intervals, reinforcing the substantial evidence supporting MDT as a primary approach.
We have revised the conclusion to clearly state that MDT alone provides excellent short- and long-term outcomes and that the addition of ADT should be carefully individualized based on patient-specific factors rather than routinely recommended. We also highlight the need for future research to refine patient selection criteria and explore biomarker-driven approaches to determine which patients may benefit from systemic therapy.
These revisions directly address your concern that the initial manuscript appeared to support the routine use of ADT in combination with MDT. We believe the updated version now presents a more data-driven and balanced perspective, emphasizing MDT as a strong standalone option while acknowledging that select patients may still benefit from combination therapy under specific conditions.
Reviewer 2 Report
Comments and Suggestions for Authors
Title - precisely depicting the focus of the study - No remarks
Abstract - concise review of the keypoints of the manuscript - No remarks
Introduction - deep analysis on the theory of oligo-met state - the dilemma of it being a specific stage of the neoplastic process or mere an early stage of metastatic cancer - No remarks
Material and methods - detailed description of the protocol of the study - PRISMA analysis, highly appropriate usage of nonparametric approach Combescure to assess compound survival probability in several single arm studies, subjected to systematic review - No remarks
Results - in-depth analysis of MDT alone vs MDT + ADT regarding DPFS, time to treatment escalation and OS. The results are diverse and substantiate the core authors idea that further sub-stratification and personalization is needed to optimize treatment in oligo-met stage of PC
Author Response
Reviewer #2:
Title - precisely depicting the focus of the study - No remarks
Thank you for your feedback. We appreciate your positive evaluation of the title and have kept it unchanged, as it accurately reflects the scope and focus of our study
Abstract - concise review of the key points of the manuscript - No remarks
Thank you for your positive evaluation of the abstract. Based on Reviewer 1's feedback, we have made some modifications to ensure that the abstract accurately reflects the key findings and conclusions of the manuscript. These revisions clarify the impact of MDT alone versus MDT + ADT on survival outcomes and emphasize the individualized approach to treatment decisions.
Introduction - deep analysis of the theory of oligo-met state - the dilemma of it being a specific stage of the neoplastic process or mere an early stage of metastatic cancer - No remarks
Thank you for your positive evaluation of the introduction. We appreciate your recognition of the in-depth analysis of the oligometastatic state, particularly its distinction as a specific stage of the neoplastic process versus an early phase of metastatic disease.
Material and methods - detailed description of the protocol of the study - PRISMA analysis, highly appropriate usage of nonparametric approach Combescure to assess compound survival probability in several single arm studies, subjected to systematic review - No remarks
Thank you for your positive evaluation of our methodology. We appreciate your recognition of the PRISMA framework and the use of the nonparametric Combescure approach to assess compound survival probability across multiple single-arm studies.
Results - in-depth analysis of MDT alone vs MDT + ADT regarding DPFS, time to treatment escalation and OS. The results are diverse and substantiate the core authors idea that further sub-stratification and personalization is needed to optimize treatment in oligo-met stage of PC
Thank you for your positive evaluation of our results. We appreciate your recognition that our findings substantiate the need for further sub-stratification and personalization to optimize treatment in the oligometastatic stage of prostate cancer.
While the results section remains largely unchanged, we have refined parts of the discussion and conclusion to ensure that our interpretation aligns with the presented data and to emphasize the role of biomarker-driven approaches in patient selection
Reviewer 3 Report
Comments and Suggestions for Authors
This manuscript reviews the role of metastasis-directed therapy (MDT) with or without hormonal therapy in oligometastatic prostate cancer (OMPC), analyzing 18 studies involving 1058 patients. It finds that combining MDT with hormonal therapy significantly improves 2- and 4-year disease progression-free survival (DPFS) compared to MDT alone in castration-sensitive OMPC, though no significant difference in overall survival (OS) was observed. The study highlights the need for biomarker-driven approaches to optimize patient selection and treatment strategies for MDT in a multidisciplinary setting. The manuscript is well-organized. However, several minor points might require attention.
- Although the authors provided a summary on metastasis-directed therapy in oligometastatic prostate cancer covering the period from 2014 to January 2024, it is recommended to update the review to include data until December 2024.
- It is unclear why “Figure 1s” was included in the main text.
- The caption titles of Figures 2 and 3 are duplicated in the manuscript.
- It is recommended to discuss the limitations of the current study in the discussion section.
- It is advisable to include the “Abbreviation” section in the manuscript.
Author Response
Reviewer #3:
This manuscript reviews the role of metastasis-directed therapy (MDT) with or without hormonal therapy in oligometastatic prostate cancer (OMPC), analyzing 18 studies involving 1058 patients. It finds that combining MDT with hormonal therapy significantly improves 2- and 4-year disease progression-free survival (DPFS) compared to MDT alone in castration-sensitive OMPC, though no significant difference in overall survival (OS) was observed. The study highlights the need for biomarker-driven approaches to optimize patient selection and treatment strategies for MDT in a multidisciplinary setting. The manuscript is well-organized.
Thank you for your positive evaluation of our manuscript. We appreciate your recognition of the study’s organization and its contribution to understanding the role of MDT with or without hormonal therapy in oligometastatic prostate cancer.
We are pleased that our analysis of DPFS, OS, and the need for biomarker-driven approaches was well received.
Although the authors provided a summary on metastasis-directed therapy in oligometastatic prostate cancer covering the period from 2014 to January 2024, it is recommended that the review be updated to include data until December 2024.
Thank you for your observation regarding the timeframe of our review. We want to clarify that our analysis included data up to December 2024, but due to an oversight, it was incorrectly stated as January 2024 in the manuscript. We have now corrected this error to accurately reflect that the review covers the full period until December 2024.
We appreciate your attention to detail and the opportunity to ensure accuracy in our work.
It is unclear why “Figure 1s” was included in the main text.
Thank you for your observation regarding Figure 1s. We initially intended to include it in the supplementary material rather than the main text. However, if you believe it adds significant value to the primary discussion, we are open to including it in the main manuscript and adjusting the figure numbering accordingly.
The caption titles of Figures 2 and 3 are duplicated in the manuscript.
Thank you for pointing out the duplication of the caption titles in Figures 2 and 3. We have revised these figures in the manuscript and ensured the captions are correctly formatted and no longer duplicated.
It is recommended to discuss the limitations of the current study in the discussion section.
Thank you for suggesting that we discuss our study's limitations. We have now added a dedicated Limitations section within the discussion to acknowledge key constraints, including the heterogeneity of the included studies, the lack of OS as a primary endpoint in many trials, potential selection biases, and the absence of biomarker-driven patient stratification. These additions ensure transparency in the interpretation of our findings and highlight areas for future research.
It is advisable to include the “Abbreviation” section in the manuscript.
Thank you for your suggestion to include an abbreviation section. We have added a list of abbreviations to improve readability and ensure clarity for readers unfamiliar with specific terminology.
Round 2
Reviewer 1 Report
Comments and Suggestions for Authors
Authors have provided thoughtful and thorough responses to the referee's comments. I applaud their work and do not identify any further outstanding issues.
Author Response
We sincerely thank the reviewer for the positive feedback and for their thoughtful evaluation of our work